# P2X7 Receptor in Dendritic Cells and Macrophages: Implications in Antigen Presentation and T Lymphocyte Activation

**DOI:** 10.3390/ijms25052495

**Published:** 2024-02-21

**Authors:** Claudio Acuña-Castillo, Alejandro Escobar, Moira García-Gómez, Vivienne C. Bachelet, Juan Pablo Huidobro-Toro, Daniela Sauma, Carlos Barrera-Avalos

**Affiliations:** 1Departamento de Biología, Facultad de Química y Biología, Universidad de Santiago de Chile, Santiago 9160000, Chile; juan.garcia-huidobro@usach.cl; 2Centro de Biotecnología Acuícola, Facultad de Química y Biología, Universidad de Santiago de Chile, Santiago 9160000, Chile; 3Laboratorio Biología Celular y Molecular, Instituto de Investigación en Ciencias Odontológicas, Facultad de Odontología, Universidad de Chile, Santiago 8380000, Chile; janodvm@gmail.com; 4Departamento de Biología, Facultad de Ciencias, Universidad de Chile, Santiago 7800003, Chile; moira.garcia@ug.uchile.cl (M.G.-G.); dsauma@uchile.cl (D.S.); 5Escuela de Medicina, Facultad de Ciencias Médicas, Universidad de Santiago de Chile, Santiago 9160000, Chile; vivienne.bachelet@usach.cl; 6Centro Ciencia & Vida, Av. Del Valle Norte 725, Huechuraba 8580000, Chile

**Keywords:** P2X7 receptor, antigen presentation, purinergic signaling, T cell activation

## Abstract

The P2X7 receptor, a member of the P2X purinergic receptor family, is a non-selective ion channel. Over the years, it has been associated with various biological functions, from modulating to regulating inflammation. However, its emerging role in antigen presentation has captured the scientific community’s attention. This function is essential for the immune system to identify and respond to external threats, such as pathogens and tumor cells, through T lymphocytes. New studies show that the P2X7 receptor is crucial for controlling how antigens are presented and how T cells are activated. These studies focus on antigen-presenting cells, like dendritic cells and macrophages. This review examines how the P2X7 receptor interferes with effective antigen presentation and activates T cells and discusses the fundamental mechanisms that can affect the immune response. Understanding these P2X7-mediated processes in great detail opens up exciting opportunities to create new immunological therapies.

## 1. Introduction

The immune system is an intricate network of cells, tissues, and molecules that acts as the body’s primary defense against pathogenic invasions and internal challenges, such as tumor development. Central to its function is the ability of the immune system to recognize and respond to specific antigens, a process that requires a coordinated interaction between various cells and molecules. Antigen-presenting cells (APCs), including dendritic cells (DCs) and macrophages (Mφ), play a pivotal role in the immune response. These cells capture antigens, process them, and display antigen fragments on their surfaces. This presentation occurs in the context of Major Histocompatibility Complex (MHC) molecules. These antigen-MHC complexes are recognized by T lymphocytes, leading to their activation and the generation of a specific immune response [1,2].

Within this framework, the P2X7 receptor (P2X7R) has emerged as a critical player in regulating antigen presentation and T cell activation. Although it was initially identified for its role in purinergic signaling and the modulation of inflammatory responses [3,4], recent research has revealed more profound functions of the P2X7 receptor in immunity. This receptor has been shown to influence the process and effectiveness of APCs, especially dendritic cells and macrophages [5,6].

In this review, we aim to explore the most recent and influential research that underscores the central role of the P2X7R in antigen presentation and subsequent T-cell activation. We will delve into the molecular and cellular biology underlying the function of the receptor, placing particular emphasis on its synergy with APCs. This interaction is crucial to understanding how the P2X7 receptor can influence immunological, autoimmune, and oncological diseases. In addition, we will address genetic variations in the receptor, such as mutations and polymorphisms, and how these can modulate the immune response, offering a broader perspective on its clinical and therapeutic relevance.

## 2. An Overview of the Biology and Function of the P2X7 Receptor

The P2X family of purinergic receptors comprises a group of membrane ionotropic receptors activated in response to extracellular ATP. Seven subtypes of P2X receptors have been identified, designated P2X1 to P2X7. The P2X7 receptor has attracted significant interest due to its role in various physiological and pathological functions. At the molecular level, P2X7R is a homotrimer, a non-selective cation channel activated by extracellular ATP. Each subunit comprises 595 amino acids, with two transmembrane domains and an extracellular loop (282 aa) containing the ATP-binding site. The intracellular N-terminal domain is short (26 aa) and, in contrast to other P2X receptor members, possesses an extended C-terminal domain (239 aa) [7,8].

One of the most distinctive features of the P2X7 receptor is its ability to form a large pore in the cell membrane when activated by high concentrations of ATP, allowing the passage of large molecules. In the context of immunity and inflammation, the P2X7 receptor plays a leading role. Its activation can induce the activation of the inflammasome, an intracellular protein complex that, once activated, releases proinflammatory cytokines, such as interleukin-1β (IL-1β) and interleukin-18 (IL-18) [9,10]. In this sense, the P2X7 receptor plays a significant role in several processes, including regulating biological barriers such as the blood–brain barrier and the blood–retinal barrier, where its activation can affect the permeability and integrity of these barriers, potentially facilitating the entry of immune cells and substances proinflammatory to neural tissue [11,12]. The chronic activation of P2X7R in neurodegenerative diseases contributes to inflammation and cell death by releasing proinflammatory cytokines, and its influence on the blood–brain barrier makes it a critical therapeutic target in neurology [13].

Furthermore, the P2X7 receptor is involved in the modulation of inflammatory responses since its activation can lead to cell apoptosis and the production of reactive oxygen species [14,15]. The P2X7 receptor, known for its role in APCs, like Mφ and DCs **[4,16]**, also modulates the activation of B lymphocytes. This modulation is carried out through intracellular signaling, which triggers the activation and proliferation of B lymphocytes. The P2X7 receptor facilitates this by influencing intracellular calcium levels and releasing the growth factors and cytokines necessary for the immune response of B lymphocytes [17].

### Pore Formation

Activation of the P2X7 receptor by extracellular ATP triggers a series of intracellular events that can culminate in the release of pro-inflammatory cytokines. When high concentrations of ATP (1–5 mM) and overstimulation occur (≥5 min), the P2X7 receptor forms a large pore in the plasma membrane, allowing the passage of molecules up to 900 Da, such as YO-PRO-1 [18,19]. This pore formation is closely related to the carboxy-terminal cytoplasmic domain of P2X7, which is essential for its formation [20,21]. Any alteration in this region, whether by deletion or single nucleotide polymorphisms (SNPs) in the C-terminal-P2X7R, abolishes the ability to activate the macropore [18,22].

The main evidence suggests that pannexin pores play a critical role in the pore-like activity of P2X7R. For example, pannexin-1 antagonists and anti-pannexin-1 iRNA reduce pore formation. Although there is a physical interaction between P2X7R and Pannexin-1, the specific details of this interaction have not yet been fully elucidated [23]. Some studies suggest that direct phosphorylation at Y308 in Pannexin 1 by Src family kinases could produce a conformational change that opens the Pannexin 1 pore [24]. These kinases, such as calcium/calmodulin-dependent protein kinase II (CaMKII), can be activated by P2X7R [25]. Still, the role of Pannexin 1 in P2X7R pore formation remains a topic of debate, because other studies have shown that the membrane pore may be an intrinsic activity of the P2X7 receptor. Karasawa et al. showed that panda P2X7 (pdP2X7), when purified and reconstituted in liposomes, can form an inherent dye-permeable pore independent of its C-terminal domain. Furthermore, they discovered that pore formation is influenced by the lipid composition of the membrane, particularly phosphatidylglycerol, and sphingomyelin, and is inhibited by cholesterol. These findings suggest that P2X7R constitutes the pore and that its opening is influenced by the lipid composition of the membrane [26]. On the other hand, the formation of the macropore due to P2X7 activation in mouse Pannexin-deficient macrophages still leads to pore dilation (as measured by YO-PRO-1 influx) [27] It has been suggested that other agonists, nucleotides such as NAD, activate the P2X7R. However, whether NAD+ is a true P2X7R agonist or lowers the ATP activation threshold is not entirely clear [28]. These studies provide new insights into P2X7R function.

## 3. Non-Canonical P2X7R Functions: Another Perspective

### Intracellular Signaling and Phospholipase Activation

The P2X7 receptor has been linked to various non-canonical (unconventional) functions. For example, it has been shown that this receptor can activate multiple intracellular signaling pathways. One of the first studies was carried out in 1992, where phospholipase D (PLD) activity in murine BAC1.2F5 macrophages after exposure to ATP and BzATP agonists was described [29]. Subsequently, research carried out by Wiley et al. and Dubyak revealed that the receptor could trigger specific signaling pathways, particularly PLD activation. While Wiley’s study focused on mouse lymphocytes and macrophages [30], Humphreys and Dubyak’s work was performed on human monocytic THP-1 cells [31]. In addition to PLD, it has been observed that P2X7R can activate other signaling pathways, such as phospholipase 2A (PLA2) and phospholipase C (PLC) [32]. Barbieri et al. demonstrated that ATP and BzATP activate p38 and the MAPK pathway in astrocytes and P2X7R-transfected HEK293 cells [33]. In parallel, activation of the P2X7 receptor is associated with stimulating various kinases, such as MAPKs [34,35,36], which can initiate downstream signaling. Using the HEK293 cell line, Amstrup and Novak pointed out that P2X7R requires the N-terminal segment for its intracellular signaling function to activate the ERK pathway [37].

Activation of the P2X7 receptor allows for a high flow of cations, such as calcium (Ca^2+^), into the cytoplasm. This increase in intracellular calcium concentrations may directly activate calcium-sensitive PLA2 in liver cells [38] and enhance PLC activity [39]. Furthermore, it has been proposed that the P2X7 receptor may activate G proteins via PLC [40]. Although progress has been made in understanding how the P2X7 receptor regulates these enzymes, the exact mechanisms and specific pathways may vary depending on the cellular and physiological contexts, underscoring the complexity and versatility of this receptor in cellular signaling.

A model proposed by Garcia-Marcos et al. suggests that the P2X7 receptor may act as a nonselective ion channel and lead to pore formation or transduce signals, depending on its location on the plasma membrane. According to this model, the receptor is distributed between lipid rafts and non-raft-associated membrane regions. In the latter, P2X7R can form homotrimers in the presence of ATP and function as an ion channel. However, in lipid rafts, the receptor remains in its monomeric conformation and does not act as a channel but instead activates intracellular signaling pathways [41]. Despite advances in understanding this receptor, there are still unanswered questions about its exact arrangement on the membrane and how this location and ATP concentration influence the cellular responses generated (Figure 1).

In addition to the non-canonical functions described above, the P2X7 receptor is essential in several cellular processes that change and rearrange membranes [42]. A prominent example is its role in cell fusion and plasma membrane movement (for example, due to the activation of phospholipases described above). In vitro experiments have shown that cells like macrophages and HEK293 cells that have a lot of P2X7R can spontaneously fuse [43,44,45,46]. Furthermore, the inhibition of P2X7R prevents the formation of giant multinucleated osteoclasts [47]. In the J774 murine macrophage model, the P2X7 receptor tends to localize to sites of cell–cell contact [48]. For both activities, the role of signaling has been described as important in the mechanisms involved in membrane fusion. In this sense, neutral sphingomyelinases PLC and PLD, activated by intracellular signaling, can produce phospholipids that facilitate the conversion of membrane topology for cell–cell fusion by catalyzing the conversion of sphingomyelin in ceramide, phosphatidylinositol bisphosphate (PIP2) in diacylglycerol (DAG) and phosphatidylcholine in phosphatidic acid, respectively, which can curve the membrane (revised in [49]).

The P2X7 receptor is typically known to be activated by ATP binding, but it is also significant for immunity and cell response even when ATP is not present. One of the least explored functions of P2X7R is its involvement in phagocytosis. It has been observed that human monocytes, in the absence of ATP, favor the phagocytosis of non-opsonized particles and bacteria via P2X7R [50]. The use of P2X7R antagonists suppresses ethidium uptake but not phagocytosis in human peripheral blood mononuclear cells (PBMCs) in serum-free conditions [51], suggesting that this activity of P2X7R is independent of membrane pore formation and ATP activation.

Gu et al. (2011) demonstrated that the P2X7 receptor facilitates phagocytosis of apoptotic cells by macrophages in the absence of serum, with ATP inhibiting this process, and highlighted the importance of the receptor’s disulfide bonds in the recognition and uptake of these cells [52]. This reinforces the idea that P2X7R, in its non-activated state, acts as a scavenger receptor. Since serum can stop P2X7R-mediated phagocytosis, it does not play a major role in tissues where plasma proteins are high in concentration. However, intense phagocytosis by monocytes and macrophages has been reported in human cerebrospinal fluid with very few serum glycoproteins [50]. It is important to note that there may be a link between P2X7R’s ability to promote phagocytosis and its connection with the cytoskeleton since ATP binding to the receptor separates it from β-actin [53,54] (Figure 2).

The discovery of unconventional functions of the P2X7R receptor, such as its role in cell fusion and membrane reorganization, could reveal crucial aspects of antigen presentation in the immune system. The ability of P2X7R to influence cellular dynamics can improve the efficiency with which antigen-presenting cells, such as macrophages and dendritic cells, acquire and present antigens to T cells.

## 4. P2X7 in Antigenic Presentation and T-Cell Activation by Dendritic Cells, an Undervalued Implication

Antigen presentation is a crucial process in the immune system that allows for the detection and adequate response to pathogens. Specialized cells like DCs and Mφ capture and process antigens before presenting them to T cells. This antigen presentation is carried out by MHC molecules, which expose fragments of antigens on the cell surface to be recognized by T cell receptors. Specific recognition of antigens allows T cells to activate and trigger an immune response directly or by activating other immune cells. This means that antigen presentation is an essential way for the body to protect itself from many types of threats, including bacterial and viral infections, and even tumor cells [55,56]. For the full activation of T cells, costimulatory molecules, such as CD80 and CD86, interact with receptors on T cells, such as CD28, providing the “second signal” necessary for optimal activation. T cells may become anergic in the absence of this costimulation, resulting in tolerance or lack of response to the antigen, even if the antigen is adequately presented [57]. Researchers have used this process to create immunotherapeutic strategies and vaccines against specific diseases [58,59].

Recent evidence suggests the role of P2X7 during antigen presentation. In this line, Di Virgilio’s group was the first to study how the P2X7 receptor is expressed in DCs. They found that follicular dendritic cells and Langerhans cells expressed high levels of this receptor. Interestingly, they also found an indirect link between P2X7R and antigen presentation for the first time by demonstrating that T cell activation decreased when the irreversible antagonist oxidized ATP (oATP) was used to block P2X7R in DCs. Intriguingly, P2X7R has been shown to modulate the expression of costimulatory molecules ex vivo [60]. In a study by Sala et al. in 2001, it was shown that chronic stimulation with low and non-cytotoxic doses of ATP (250 μM) enhances the expression of CD54, CD80, CD86, and CD83 in DCs derived from peripheral blood monocytes, also expanding their capacity to promote the proliferation of naïve allogeneic T lymphocytes [61]. Consistent with these findings, Furuta et al. in 2023 showed that high concentrations of extracellular ATP (1 mM) enhanced the expression of molecules related to antigen presentation, such as MHC-I, MHC-II, and costimulatory molecules, such as CD80 and CD86. Furthermore, an increase in the production of interleukins, such as IL-1β, IL-6, IL-12, and IL-10, was observed in mouse bone marrow-derived DCs (BMDCs), which mediated T cell activation by inflammasome activation [62]. These findings highlight the relevance of P2X7R modulation in DCs.

In a study carried out in 2019 by Yu’s team to elucidate part of the mechanism through which P2X7R regulates DC function, it was evidenced that the NF-κB (p65) pathway plays a fundamental role in the maturation of BMDCs after activation of the P2X7 receptor by ATP. This activation leads to the induction of the costimulatory proteins CD80 and CD86 and the release of IFN-γ and IL-12 in DCs [63]. However, it is notable that, despite the positive effects of ATP on DC maturation, the combination of ATP and LPS (lipopolysaccharide) distorts this maturation process. In the presence of both ATP and LPS, an inhibition in the production of IL-1β, TNF-α, IL-6, and IL-12 is observed due to chronic stimulation compared to treatment with LPS alone. These findings suggest that extracellular ATP can act as an essential regulatory signal against molecules or pathogens, modulating the production of IL-12 in DCs, as extensively reviewed previously [64]. This modulation may be a strategy for the body to prevent excessive and potentially harmful immune responses.

In research carried out by Baroja-Mazo et al. in 2013, the levels of the MHC-I molecule decreased on the surface of DCs and in bone marrow-derived macrophages after activation of the P2X7 receptor with ATP [65]. These findings suggest that the activation of this receptor can have an inhibitory effect on the response mediated by T lymphocytes. The authors proposed that this decrease in MHC-I complexes could be due to the release of MHC-I through vesicle shedding, as P2X7R is known to induce the release of vesicles containing MHC-II protein [66]. It is relevant to note that the concentration of ATP used in this study was 5 mM for 30 min, which differs from the concentrations used in the previously described research. In biomedical research, the millimolar concentrations of ATP used in vitro to activate the P2X7 receptor contrast with the micromolar concentrations found in vivo, which is due to several key reasons. First, the P2X7 receptor is characterized by requiring higher concentrations of ATP for its activation compared to other P2X receptors, possibly due to its unique molecular structure and role in cellular signaling [67]. In vitro studies employ these high concentrations to ensure receptor activation and to study its effects in a controlled manner. In contrast, in vivo, the availability and concentration of extracellular ATP are regulated by multiple factors, such as ATP-degrading enzymes and tissue fluid dynamics. Moreover, although the levels of extracellular ATP (eATP) can increase in pathological conditions, such as tumors or inflammation, they rarely reach millimolar levels. However, it has been observed that eATP concentrations of 100–300 μM, typical in damaged tissues or under cellular stress, are sufficient to activate the P2X7 receptor, even apoptosis [68]. This indicates that although high concentrations are used in vitro to study the full range of receptor responses, in vivo activation and physiological function of the receptor do not necessarily require such elevated levels [69] (Figure 3).

## 5. P2X7 Receptor and Antigen Cross-Dressing in DCs: A New Way to Present Antigens and Activate T-Cells

The P2X7R receptor plays a fundamental role in antigen presentation and the activation of T cells through a process known as antigen cross-dressing. “Cross-dressing” is a process through which APCs, like DCs, can take up preformed peptide/MHC complexes from other cells’ membranes and put them on their surfaces without antigen processing [70]. A study carried out by Barrera-Avalos et al. (2021) reported that P2X7R expression is essential for the transfer of preformed functional complexes of peptide from Ovalbumin (OVA), SIINFEKL_OVA257-264_/MHC-I, from the surface of extracellular vesicles (EVs) to the surface of BMDCs, and conventional CD8^+^ DCs (cDC1). The peptide/MHC-I complex can be transferred to the surface of DCs in a fully functional way that can activate CD8^+^ T lymphocytes [71]. To date, it is unknown how P2X7R promotes antigen cross-dressing in DCs. However, it is postulated that it could be through its previously described membrane-fusogenic activity by interacting with another P2X7R on its cell membrane counterpart or activating downstream signaling after the recognition of the EVs by the DCs. It is interesting to note that this mechanism of antigen presentation has been reported to be essential in the resolution of viral infections and the immune response against cancer [72,73].

The possible role of P2X7R in the induction of cross-dressing is indirectly supported by some clinical conditions, for example, in graft versus host disease (GVHD). This is a common complication after bone marrow transplants, where the donor’s T cells recognize and attack the recipient’s tissues as foreign. Markey et al. (2014) [74] described how the phenomenon of “cross-dressing”, in which murine donor dendritic cells adopt MHC molecules from the recipient, contributes to the formation of the immunological synapse, optimizing immune responses to antigens from the recipient presented indirectly. This could worsen GVHD by boosting the immune response against recipient tissues. While cross-dressing has been directly linked to GVHD, the P2X7 receptor has been described as actively influencing the development of the disease. In 2010, Wilhelm et al. reported that ATP contributed to GVHD by showing that ATP levels rose during GVHD in both mice and humans and that ATP stimulation boosts the immune response, mostly through the P2X7R receptor. In contrast, P2X7R-deficient animals resulted in increased STAT5 phosphorylation and FoxP3 expression in CD4^+^ T cells, thereby promoting immunological tolerance. Furthermore, GVHD can be mitigated by ATP neutralization or P2X7R modification, suggesting a therapeutic potential to reduce the severity of GVHD without requiring intensive immunosuppression in a mechanism that would decrease receptor-mediated cross-dressing [75]. Similarly, P2X7R antagonism reduces the progression of clinical GVHD in a humanized mouse model [76,77].

In addition, Sluyter’s group has also made significant contributions to determining the role that purinergic receptors play in GVHD [78]. An interesting study by Cuthbertson et al (2021) demonstrates that although there is a correlation between P2X7 receptor activity and P2RX7 genotype in human leukocytes (gain or loss of receptor function), this relationship does not have a significant impact on the development of the disease in a humanized mouse model [79]. This means that, while genetic differences were found to cause differences in P2X7 activity in humans, these differences were not recapitulated in the experimental model used to see changes in the severity or presence of GVHD. This finding is relevant because it suggests that although the P2X7 receptor and its genetic variants may play a role in the immune response and related diseases, they are not necessarily key determinants in the development of GVHD in the context studied (Figure 4). They corroborate that the mechanisms underlying complex diseases, such as GVHD, may be multifactorial and that a single receptor or pathway may not be sufficient to predict or modify the course of the disease.

### P2X7 Receptor in Antigen Cross-Presentation by Dendritic Cells: A New Approach

In the same study by Barrera-Avalos et al. (2021), it was shown that incubating extracellular vesicles (EVs) that expressed the P2X7 receptor and endogenous OVA, the cross-presentation of OVA in BMDCs was favored compared to EVs without P2X7R. Interestingly, when P2X7-deficient BMDCs (P2X7-KO) were incubated with EVs expressing P2X7R and OVA, there was no increase in the cross-presentation of OVA [71]. This result suggested that P2X7R in both DCs and EVs or antigenic sources could favor the cross-presentation of antigens with subsequent activation of T lymphocytes. This could be because, as previously described for the P2X7 receptor, membrane fusion between EVs and DCs may occur, favoring the transfer of OVA to the cytoplasm of DCs to induce cross-presentation of antigen through the cytosolic route [80] (Figure 5). Interestingly, Giuliani et al. investigated the shedding of the P2X7 receptor into the bloodstream and its correlation with C-reactive protein (CRP) levels, an inflammation marker. It reveals that the shed P2X7 receptor (sP2X7R) was found in the blood of healthy subjects and patients, with its levels increasing in various conditions. The study suggests that sP2X7R in blood might provide a novel diagnostic approach to monitoring inflammation, as its levels significantly correlate with CRP levels, particularly in conditions like ischemia, infections, and other inflammatory diseases. The research also explores the potential sources of sP2X7R, suggesting its association with EVs like microvesicles/microparticles released from cells [81] The release of microparticles that express P2X7R due to different pathologies could, for example, be phagocytosed by DCs and cross-present the antigens carried on these microparticles to T lymphocytes in a mechanism that the P2X7 receptor could favor.

## 6. P2X7R in Macrophages: An Indirect Effect on Antigen Presentation

In macrophages, P2X7R is highly expressed and is an essential component of the innate immune reaction against pathogens, playing a crucial role in the control of pro- inflammation and anti-inflammation in an immune response [82].

One of the functions attributed to P2X7R in macrophages is related to their polarization. In 2023, Scherr et al. published a study that showed that ATP slows down the natural differentiation of human monocytes to M2 macrophages. This was linked to a dose-dependent decrease in the release of chemokine CCL18. However, ATP stimulation did not affect IL-1β concentration, indicating that ATP prevents differentiation into M2 but does not affect differentiation into M1 during an inflammatory process [83]. Importantly, M1 macrophages release the cytokines IL-12 and IL-23 and promote the differentiation of T cells into Th1 and Th17 cells, respectively, which is essential for the immune response against intracellular pathogens and inflammation [84]. In this line, the immune response against this type of pathogen could be improved by promoting macrophages to become more like the M1 phenotype through P2X7R. This is achieved by increasing T cell activation during antigen presentation by macrophages and consolidating the relationship between antigen presentation and P2X7R-mediated T cell activation in macrophages under some infections.

In 2010, Thomas and Salter showed that the expression of CD86, CD80, MHC-II, and TNF-α increases when 3 mM ATP was added to murine J774 macrophages. This effect is dependent on P2X7 [85]. This expression suggests a potential augmentation of antigen presentation capabilities and subsequent T-cell stimulation. At the same time, parts of bacteria, especially LPS, have been found to interact with the C-terminal domain of P2X7R. Such interactions modulate the receptor’s activity, increasing the production of pro-inflammatory cytokines TNF-α and IL-1β [86].

In a study conducted in 2020, Zhou and colleagues highlighted the role of the MerTK receptor in recognizing apoptotic cells within Mφ. They found that inhibiting the MerTK receptor promoted ATP accumulation within the tumor microenvironment. This extracellular ATP, in turn, can activate the P2X7R, facilitating the entry of dsDNA, although the mechanism of cGAMP entry is still unclear. This process triggers the activation of cGAMP and the STING pathway, culminating in the production of IFN-β. This series of events causes IFN-β-dependent effector T cells to become more polarized, which boosts the immune response against the tumor [87] (Figure 6). Still, the complicated link between activating P2X7R and its direct effect on antigen presentation in macrophages is a promising but not fully explored area of scientific research.

### P2X7R in Mycobacterium Tuberculosis Infections

Tuberculosis is caused by *Mycobacterium tuberculosis* (MTB), which enters the body mainly through inhalation and predominantly infects alveolar macrophages in the lungs. This bacterium has developed mechanisms to survive and multiply within macrophages, thus evading the body’s immune defense [88]. Several studies have indicated that activation of the P2X7R receptor is crucial in eradicating *M. bovis* infection in ex vivo cultures of human monocyte-derived macrophages (MDMC) [89]. ATP has also been shown to improve the killing of *M. tuberculosis* in THP-1 monocytes in vitro [90]. A study led by Santos et al. in 2013 revealed that P2X7R-deficient mice (P2X7R^−/−^) had a higher bacterial load of *M. tuberculosis* in the lung tissue and a reduction in the number of CD8^+^ T lymphocytes, in contrast to wild-type (WT) mice [91]. However, other studies have shown divergent results, where mice lacking the P2X7 receptor (P2X7^−/−^) exhibited no significant differences in lung bacterial load or interferon-gamma (IFN-γ) levels compared to WT mice. Furthermore, ex vivo antigen presentation was analogous between P2X7^−/−^ and WT mice [92]. Another analysis showed similar findings, where mortality and lung deterioration in P2X7^−/−^ mice manifested with a delay compared to WT mice [93]. Despite these discrepancies, ATP has been proposed as a therapeutic strategy to enhance the clearance of *M. tuberculosis* and promote apoptosis in infected monocytes and macrophages through the activation of P2X7R [94]. This activation could not only induce the death of the bacteria but also enhance other functions of P2X7R, such as the elevation of costimulatory molecules and the production of proinflammatory cytokines.

A promising approach to the immune response against *M. tuberculosis* lies in the apoptosis of infected cells [95], giving rise to apoptotic cell-derived extracellular vesicles (ACdEVs) carrying *M. tuberculosis* antigens (reviewed in [96]). It has been shown that ACdEVS originating from macrophages infected with mycobacteria can trigger a specific CD8^+^ T cell response in vivo [97,98,99]. One of the possible pathways of the immune response against *Mycobacterium tuberculosis* could be mediated by P2X7R since these ACdEVs would contain P2X7 receptors on their surface, favoring cross-dressing or cross-presentation of antigens as an alternative to T-cell activation never previously seen, probably involving membrane fusion as previously described (Figure 7).

## 7. P2X7 Receptor Loss of Function Polymorphisms Restrain the Immune Response

The P2X7R gene is known to have numerous polymorphisms and single nucleotide polymorphisms (SNPs). However, not all these variations have been linked to changes in response to diseases in the existing literature, but only a few have been studied in this context.

Two notable variants of P2X7R are P2X7B, which has an out-C-terminal domain, and P2X7J, which is non-functional due to the truncation of exon 7. Both P2X7R variants lack pore-forming activity [100]. This absence is significant, as it contributes to cancer progression, primarily because these variants cannot induce apoptosis in tumor cells. Another overlooked factor is its inability to facilitate adequate T-cell activation.

The most extensively studied P2X7R polymorphism is 1513A>C (Glu496Ala). This variant retains its electrophysiological characteristics but is unable to form a macropore [101,102] due to lower expression of the P2X7 receptor in vivo, as demonstrated in monocytes from patients homozygous for the Glu496Ala mutation [103]. Interestingly, the 1513A>C variant makes it harder for the host to fight off *M. tuberculosis* so the bacteria can live inside host cells [104,105]. This polymorphism is also linked to a heightened risk of pulmonary tuberculosis in Asian populations [106,107]. Furthermore, it has been associated with increased susceptibility to *M. tuberculosis* infection in various global populations [108,109,110,111,112,113]. This variant also correlates with several cancers, including B-chronic lymphocyte leukemia [114], papillary thyroid cancer [115], and breast cancer [116]. In addition, it has shown a reduced anti-tumor immune response against multiple myeloma [117].

In contrast, the SNP Pro 451 Leu, found in mice, affects macropore formation but does not alter the channel’s properties in murine thymocytes [118]. Its role in antigen presentation or infection has yet to be assessed. A 2017 study evaluated the susceptibility of the B16F10 metastatic melanoma line in two mouse strains, C57BL/6 and BALB/c. These strains differ in their MHC haplotypes; the BALB/c strain carries the P2X7R Pro451Leu SNP. The effect of this difference can be observed, for example, when B16F10 melanoma cells are administered into BALB/c mice, when they have three times more likely to develop melanoma than C57BL/6 mice. This finding underscores the importance of a functional P2X7R response, even when a tumor of one genetic profile develops in a host with a different genetic background [119]. The importance of a functional P2X7R could also be related to a decrease in the antigen presentation of the B16F10 line and a decrease in the activation of T lymphocytes, which are important processes against tumor development.

Additionally, C57BL/6 mice showed resistance to lethal encephalitis when infected with Herpes simplex 1 (HSV-1) compared to BALB/c mice [120]. This resistance is attributed to the enhanced activity of CD8^+^ and T-CD4^+^ T-lymphocytes in C57BL/6 mice. A similar difference in immune responses between the two mouse strains was observed during *Yersinia enterocolitica* infection [121]. However, it is essential to note that the varied immune responses between BALB/c and C57BL/6 mice might not solely be due to the P2X7R polymorphism.

The polymorphism 946G>A, which results in the substitution of glutamine (R307Q) for arginine in position 307 of the amino acid, causes a loss of function by switching the location of the ATP-interaction site in the extracellular domain [122], relating to a decrease in proinflammatory status. Hence, it is likely that the receptor polymorphism not only affects the ability of tumors to enter apoptosis due to the inability to generate the membrane pore but also directly affects the ability of APCs to induce T-cell activation.

Among the splice variants of the P2X7 receptor, only P2X7B maintains its activity as a channel. The other described variants, including C, E, F, and G, are non-functional [123,124]. Recent analyses that include these variants have allowed us to characterize that their differential expression correlates with the malignancy of the disease. For example, in breast cancer, the P2X7J variant was identified that lacks the second transmembrane domain and the C-terminal tail. This variant is endogenously expressed in cervical cancer cells and is responsible for cancer progression [125]. The P2X7B splice variant, associated with a lack of macropore formation, increases the proliferation of Te85 osteosarcoma [126]. This is expressed in several tumors and is correlated with metastasis, greater aggressiveness, and greater chemoresistance in leukemia and osteosarcoma [127,128].

A common point between the P2X7R splice variant and SNPs is the association with a lack of macropore formation after a stimulus. Initially, it was described that the presence of P2X7R in tumor cells results in an unfavorable prognosis in most cases. However, we only sought to determine whether this receptor was expressed in the different tumor stages without considering the presence of receptor polymorphisms [129,130,131,132]. Overall, research into the connection between P2X7R polymorphisms and diseases continues. More research is required to fully understand how these genetic differences affect both health and disease in the population. It is crucial, for example, at the level of maturation, antigen presentation, and activation of T lymphocytes by dendritic cells and macrophages, since these are critical events to initiate an effective immune response. These aspects have not yet been related to the consequences of P2X7 receptor polymorphisms.

## 8. P2X7 Receptor Gain of Function Polymorphisms Increase the Immune Response

The main role described for P2X7R linked to the activation of the functional response is associated with the induction and activation of the inflammasome, but several reports show that P2X7R SNPs and splicing variants are expressed in cells and could cause immune response disruption. Pegoraro et al. (2020) associated the C489T (H155Y) gain-of-function on macropore induction activity polymorphism with increased susceptibility to human herpes virus-6 (HHV-6) infection [133]. Furthermore, it suggested that the virus’s virulence and degree of illness, in this case, depended on a high activation of the P2X7R (revised in [134]). As a result, the P2X7R can promote infection by favoring pathogen entry through the formation of membrane pores generated by activating the P2X7R. The H155Y mutation, which generates a gain of function, causes greater activation of the NLRP3 inflammasome, increasing the levels of IL-1βb, IL-18, and TNF cytokines, yielding a scenario of extreme inflammation [135]. This exacerbated inflammation has been linked to Alzheimer’s disease [136], autoimmune diseases [137], and chronic pain [138]. This background could suggest that the gain-of-function polymorphism could aggravate, for example, an infection by the recent SARS-CoV-2 virus, considering that a pro-inflammatory storm is responsible for the lung and multicellular damage associated with fibrosis [139], in an unknown aspect. These studies indicate the importance of the P2X7 receptor in generating an inflammatory state and open the possibility that these findings are related to the capacity of the receptor to favor the presentation of pathogen antigens and the activation of T lymphocytes. The effects of the different polymorphisms and splicing variants of P2X7R are summarized in Table 1.

## 9. Concluding Remarks

The P2X7R, an ionotropic receptor, is a member of the P2X family of purinergic receptors. It has been identified as playing several roles within immune cells, including the modulation and differentiation of T cells, dendritic cells, and macrophages, with a particular emphasis on inflammatory responses. The receptor’s function within antigen-presenting cells is becoming increasingly clear, identifying numerous significant functions. For instance, it has been reported that the activation of P2X7R on dendritic cells triggers the expression of costimulatory molecules and the secretion of cytokines, thereby facilitating T-cell activation. The activation of T lymphocytes against various antigens may also be attributed to a relatively novel and under-researched mechanism known as cross-dressing and cross-presentation. Moreover, other populations of antigen-presenting cells, such as macrophages, can also be modulated through P2X7R activation. This modulation can result in either the stimulation of T-cell responses or the resolution of inflammation, contingent on whether the reaction is mediated by M1 inflammatory Mφ. In addition to its association with other aspects of the physiology of presenting cells, we have summarized the reported evidence of the receptor’s role in conventional and unconventional antigen presentation related to canonical and non-canonical P2X7R functions. Direct evidence implicates the receptor’s role in cross-dressing in response to MTB. Furthermore, evidence-based polymorphism supports the possible role of the receptor in unconventional antigen presentation.

The modulation of the P2X7R has excellent therapeutic potential in treating infectious diseases and cancer, but more research is required to develop effective and safe therapeutic strategies. These strategies include using P2X7R agonists or antagonists, or modulation of ATP release to tune P2X7R activation and optimize the immune response. However, it is essential to emphasize that despite the increasing awareness of the role of P2X7 in human diseases, no P2X7-targeted drugs have been translated to clinical use. This suggests that more research is required to fully understand this receptor’s basic molecular and physiological properties and the intracellular pathways its activation or inhibition affects. Furthermore, it is worth noting that using P2X7R agonists or antagonists, or modulation of ATP release, to tune P2X7R activation and optimize the immune response should consider the immunological context. This means that the approach should be tailored to the specific pathogen in the case of infectious diseases or to the type and stage of cancer to optimize the effectiveness of the therapeutic strategy. These findings collectively open up new avenues for developing innovative therapies that target or activate the P2X7R to enhance the immune response in various disorders, including cancer.

## Figures and Tables

**Figure 1 ijms-25-02495-f001:**
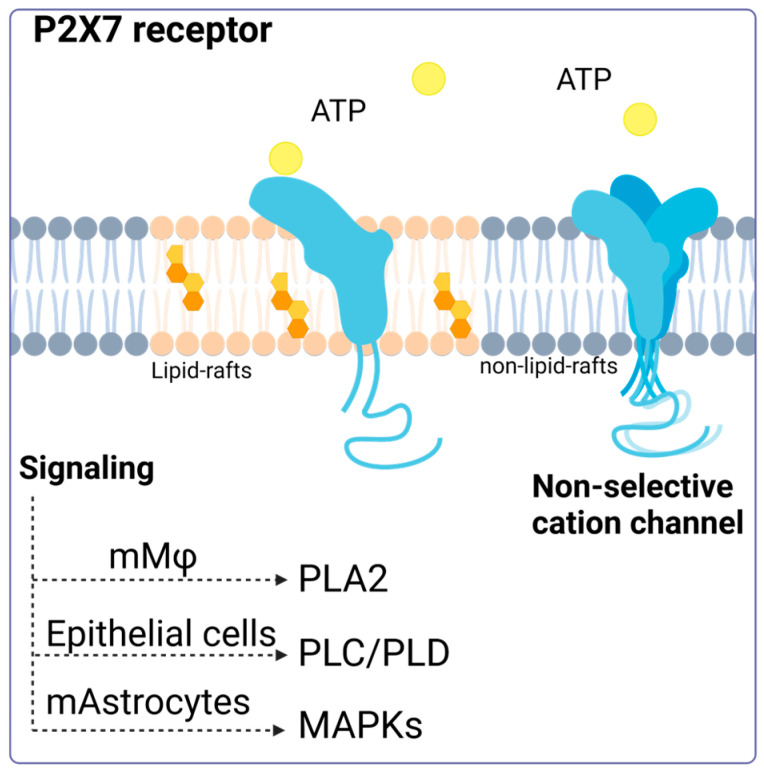
P2X7 receptor activation. Activation of the P2X7 receptor can have different consequences depending on its location (membrane modeling). The P2X7 receptor located in lipid rafts can maintain its monomeric structure, and its activation triggers signaling via phospholipase 2A (PLA2), phospholipase C, phospholipase D, and the MAP kinase pathway. Accessed on 13 January 2024, (https://www.biorender.com/).

**Figure 2 ijms-25-02495-f002:**
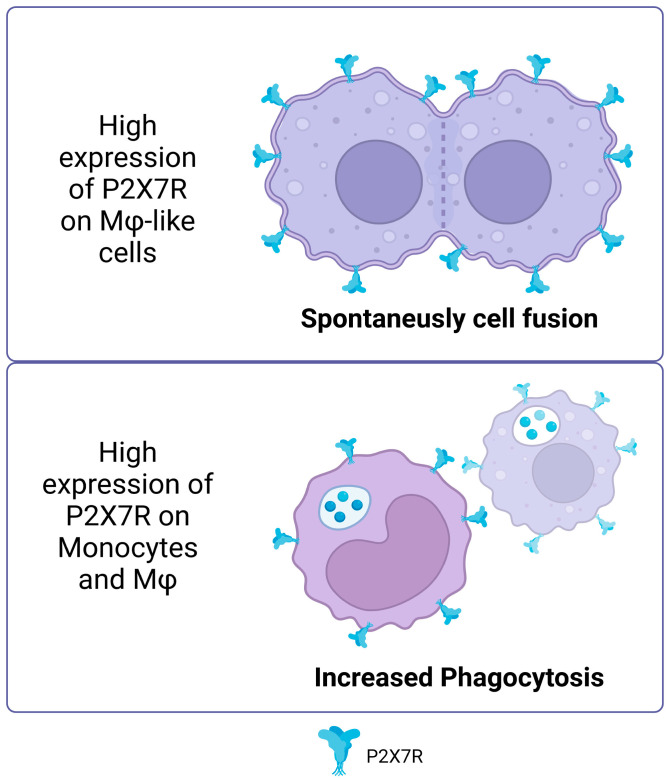
Non-canonical P2X7R functions. High expression of P2X7R on macrophage like cells promotes spontaneous cell fusion. In monocytes and macrophages, high expression of P2X7 increased the phagocytosis of non-opsonized particles. Accessed on 25 November 2023, https://www.biorender.com/.

**Figure 3 ijms-25-02495-f003:**
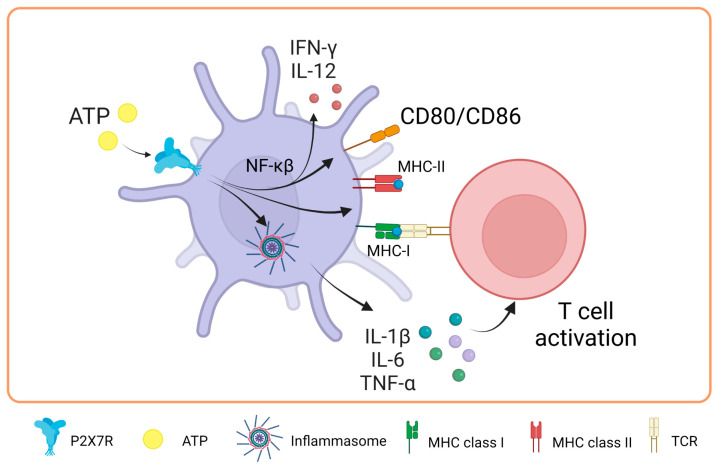
Role of P2X7 receptor in dendritic cells. In dendritic cells, the activation of P2X7 promotes the expression of major histocompatibility complex class I and II molecules and the costimulation of T-cells through the expression of CD80/CD86 by NF-κβ pathway. Moreover, it activates the inflammasome, which promotes IL-1β, IL-6 and TNF-α release favoring the T cell activation. Accessed on 24 November 2023, https://www.biorender.com.

**Figure 4 ijms-25-02495-f004:**
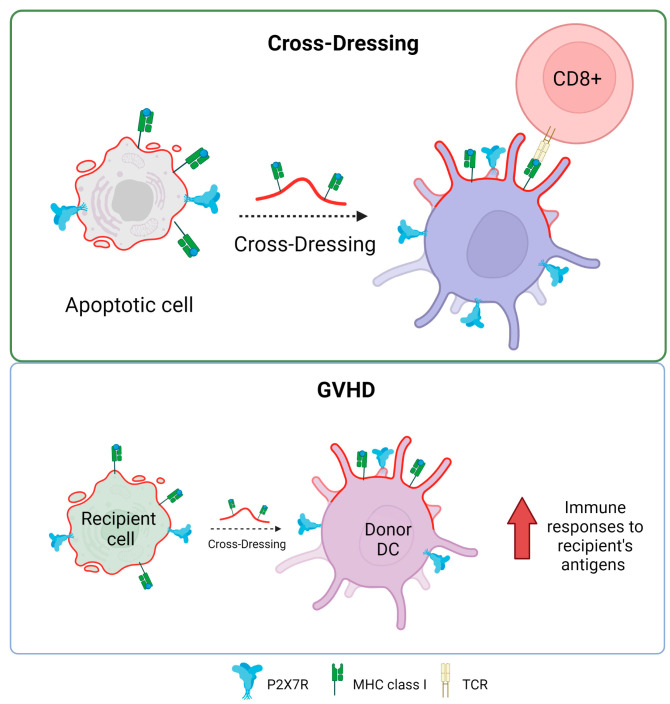
P2X7 receptor and antigen cross-dressing in DCs. The presence of P2X7 is essential for the cross-dressing process of dendritic cells and antigen presentation in MHC class I to CD8^+^ T lymphocytes. The process of cross-dressing can be involved in graft versus host disease (GVHD). Accessed on 21 November 2023, https://www.biorender.com.

**Figure 5 ijms-25-02495-f005:**
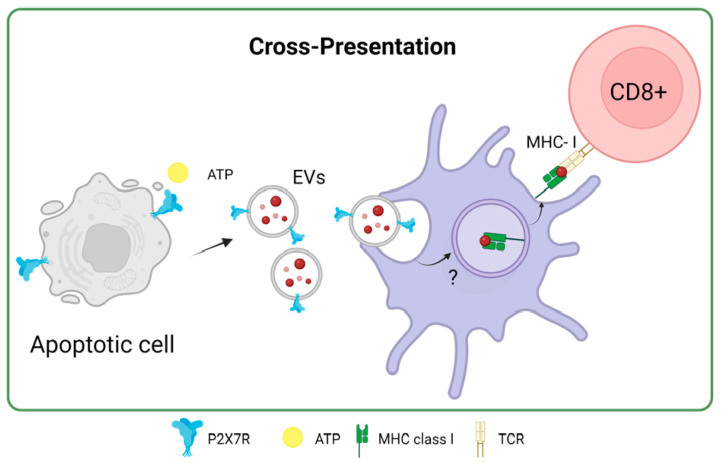
P2X7 receptor in antigen cross-presentation by dendritic cells. Accessed on 27 November 2023, https://www.biorender.com.

**Figure 6 ijms-25-02495-f006:**
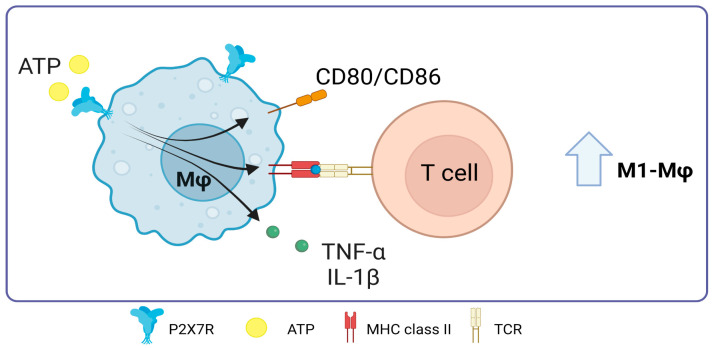
Role of P2X7 receptor in macrophages. In macrophages, the activation of P2X7 promotes the expression of major histocompatibility complex class II molecules and the costimulation of T-cells through the expression of CD80/CD86, it also promotes TNF-α release. Accessed on 28 November 2023, https://www.biorender.com.

**Figure 7 ijms-25-02495-f007:**
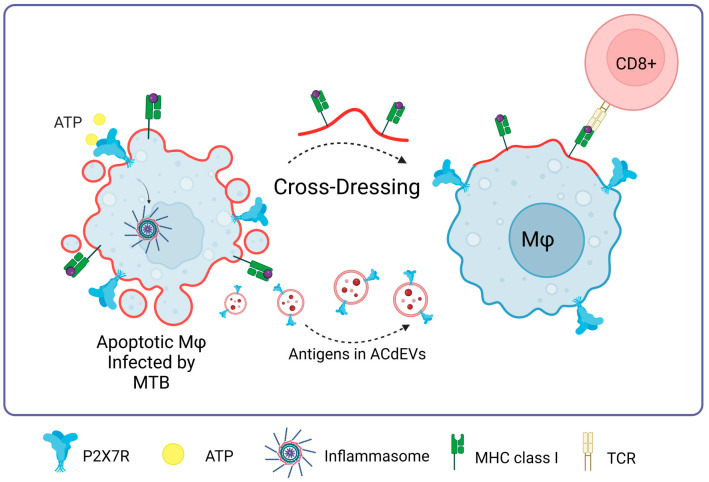
P2X7R in *Mycobacterium tuberculosis* infections. The presence of P2X7 is fundamental to the process of macrophage cross-dressing in the context of MTB and CD8^+^ T cell activation. MTB antigens contained in apoptotic cell-derived extracellular vesicles (ACdEVs) from apoptotic macrophages can activate a CD8^+^ T cell-specific response. Accessed on 26 November 2023, https://www.biorender.com.

**Table 1 ijms-25-02495-t001:** Relationship between SNPs and P2X7R variants in the responses mediated by APCs.

Mutation Studied	Effect	Reference
	**P2X7R Polymorphism**	
1513A>C (E496A) (human)	It generates a receptor that retains its electrophysiological features but cannot form a macropore.	[101,102]
	Impairs the response against MTB and allows the survival of mycobacteria within the host cells in several populations.	[104,105,108,109,110,111,112,113]
	Decreased antitumoral immune response against cancers	[104,105]
451P>L (mouse)	Affected macropore formation without channel’s properties alteration.	[118]
946G>A (R307Q) (human)	Loss of function by switching the location of the ATP-interaction site in the extracellular domain. This decreases proinflammatory status	[122]
	Increased susceptibility to human herpes virus-6 (HHV-6) infection by gain-of-function on macropore induction activity.	[133]
	Causes greater activation of the NLRP3 inflammasome, increasing the levels of IL-1βb, IL-18, and TNF cytokines.	[135]
	**Splicing variants**	
P2X7B (human)	Lack of macropore formation, correlated with metastasis and more aggressive, and even more chemoresistance in leukemia and osteosarcoma.	[127,128]
P2X7J	Lack of macropore formation.	[100]
	Variant identified in breast cancer, which lacks the second transmembrane domain and the C-terminal tail.	[125]
P2X7 C, E, F, and G (human and mouse)	non-functional.	[123,124]

## Data Availability

Not applicable.

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
