# Peer review of "P2X7 Receptor in Dendritic Cells and Macrophages: Implications in Antigen Presentation and T Lymphocyte Activation"

_ijms, 2024, doi:10.3390/ijms25052495_

Round 1
Reviewer 1 Report (New Reviewer)
Comments and Suggestions for Authors
The authors present an interesting, comprehensive and encouraging critical review on the role of the P2X7 receptor in the immune response, in particular antigen presentation.
Here are a few comments that may improve the work.
1.) P1: “… macrophages (MΦ), capture antigens, process them, and then present them on their surface …”
Rather, parts of the antigen are presented
2.) P2: “The P2X7 receptor modulates the functions of macrophages (MΦ), dendritic cells (DCs), and T lymphocyte activity …”
The P2X7 receptor also modulates B lymphocyte activity, plese cite relevant work, for example also
Pippel et al. Cell Calcium 57 (2015), 275-289.
3.) P2: “… the P2X7 receptor forms a large pore in the plasma membrane, allowing the passage of molecules up to 900 Da, such as YO-PRO-1 …”
Please cite a relevant revie, for example
Di Virgilio et al. Trends Cell Biol. 28 (2018), 392-404.
4.) P5: “Researchers have used this process to create immunotherapeutic strategies and vaccines against specific diseases.”
Please cite relevant literature.
5.) P6: “It is relevant to note that the concentration of ATP used in this study was 5 mM for 30 minutes …”
The high ATP concentrations (>1 mM) used in several investigations should be discussed, since such concentzrations will not occur extracellularly in vivo.
6.) P9: “This extracellular ATP, in turn, can activate the P2X7R, facilitating the entry of
dsDNA via the formation of membrane pores.”
This is an overinterpretation. In the related article is stated that: “The entry of tumor-cell-produced cGAMP into host cells was facilitated by the ATP-gated P2X purinoceptor 7 (P2X7) receptor (P2X7R)".
That means that the mechanism of cGAMP entry is not yet clear.
7.) P11: “This variant retains its electrophysiological characteristics but is unable to form a macropore.”
An explanation for the retained electrophysiological phenotype but a reduced biological function may be a reduced expression in vivo. See the relevant work of J. Wiley et al.
8.) P11: Reference 88 is wrongly cited, see References.
Comments on the Quality of English Languageno further comment
Author Response
Reviewer 1
The authors present an interesting, comprehensive and encouraging critical review on the role of the P2X7 receptor in the immune response, in particular antigen presentation.
Here are a few comments that may improve the work.
1.) P1: “… macrophages (MΦ), capture antigens, process them, and then present them on their surface …”
Rather, parts of the antigen are presented
Answer: Thanks for your comment. This was corrected in the manuscript based on your suggestion. (L36-40)
2.) P2: “The P2X7 receptor modulates the functions of macrophages (MΦ), dendritic cells (DCs), and T lymphocyte activity …”
The P2X7 receptor also modulates B lymphocyte activity, plese cite relevant work, for example also
Pippel et al. Cell Calcium 57 (2015), 275-289.
Answer: Thanks for your comment. This review focuses mainly on the function of the P2X7 receptor in antigen-presenting cells (APCs), specifically macrophages (Mø) and dendritic cells (DCs), and its role in the activation of T cells through antigen presentation. Although, as per your suggestion, we briefly mentioned that the P2X7 receptor also influences B cells, this aspect is not the main focus and, therefore, is not detailed in depth in our study (L81-86)
3.) P2: “… the P2X7 receptor forms a large pore in the plasma membrane, allowing the passage of molecules up to 900 Da, such as YO-PRO-1 …”
Please cite a relevant revie, for example
Di Virgilio et al. Trends Cell Biol. 28 (2018), 392-404.
Answer: Thanks for the observation. As you suggested, the relevant reference for this sentence was added to the manuscript (L92)
4.) P5: “Researchers have used this process to create immunotherapeutic strategies and vaccines against specific diseases.”
Please cite relevant literature.
Answer: The reference for this sentence was added to the manuscript (L210-212).
5.) P6: “It is relevant to note that the concentration of ATP used in this study was 5 mM for 30 minutes …”
The high ATP concentrations (>1 mM) used in several investigations should be discussed, since such concentzrations will not occur extracellularly in vivo.
Answer: Thank you for your observation. This was discussed in the manuscript in L250-264.
6.) P9: “This extracellular ATP, in turn, can activate the P2X7R, facilitating the entry of
dsDNA via the formation of membrane pores.”
This is an overinterpretation. In the related article is stated that: “The entry of tumor-cell-produced cGAMP into host cells was facilitated by the ATP-gated P2X purinoceptor 7 (P2X7) receptor (P2X7R)".
That means that the mechanism of cGAMP entry is not yet clear.
Answer: Thanks for the comment, this overinterpretation was corrected (L381)
7.) P11: “This variant retains its electrophysiological characteristics but is unable to form a macropore.”
An explanation for the retained electrophysiological phenotype but a reduced biological function may be a reduced expression in vivo. See the relevant work of J. Wiley et al.
Answer: A relevant article by Wiley et al. about the low expression of P2X7 in patients with the 1513 A > C polymorphism was cited (L440-441)
8.) P11: Reference 88 is wrongly cited, see References.
Answer: The polymorphism 1513 A > C generates E496A change. This was modified to Glu496Ala (L438). The reference 88 its ok.

Reviewer 2 Report (New Reviewer)
Comments and Suggestions for Authors
The authors of this review summarized recent literature on the role of P2X7 in immune responses. They focused on the involvement of P2X7 in the antigenic presentation by dendritic cells and macrophages, the activation of T-cells, and the effect of loss or gain of functional polymorphisms on the modulation of immune responses. The authors showed that many P2X7 activities are still waiting to be explained, and concluded that the link between the activation of P2X7 and its direct effect on antigen presentation is a promising area of scientific research that may lead to new immunological therapy mediated by the P2X7 receptor. It is a highly informative, actual and excellently written overview that I don't have any great comments on.
I only have one suggestion I'm giving authors to consider: recent studies have shown that the P2X7 receptor can be released into the circulation and that the level in the blood increases with various diseases (Giuliani et al., Front Immunol, 2019. 606 10: pp. 793). The authors could comment on possible changes in the level of P2X7 in the blood during inflammatory processes.
Minor Comments
Some abbreviations are introduced repeatedly in the text (such as APC and DCs); they should only be mentioned once, when they are used for the first time.
Page 1, line 8 from the bottom: : „MΦ” is this abbreviation correct? It seems not to be used systematically.
Page 2, line 4: abbreviation „P2X7R“ has not been introduced and is also not systematically used
Page 2, line 6: abbreviation APCs aleardy exists
Page 2, line 18 from the bottom: abbreviation DCs aleardy exists
Page 2, line 8 from the bottom: “Strong evidence suggests that pannexin pores play a critical role in the pore-like activity of P2X7R. “ The authors should mention also other possibilities. More recent studies demonstrated using whole-cell recordings that NMDG+-mediated currents occur without delay after agonist exposure (Harkat, Peverini et al. 2017, Pippel, Stolz et al. 2017), for example. Measurement of the dye uptake activity of purified panda P2X7 receptor reconstituted into liposomes showed that P2X7 forms a dye-permeable pore in the absence of other cellular components but is dependent on the lipid composition of the membrane (Karasawa, Michalski et al. 2017).
Page 3, line 4: “el-Moatassim C and Dubyak GR identified…“ The names of the authors mentioned in the main text should be without abbreviations of their first namest. The same problem is on page 4, line 7 from the bottom and in other places
Page 4, line 2: “In addition to its normal functions,,,,“ What is “normal function”? Please, specify.
Page 5, line 8 from the bottom: abbreviation „MHC“ already exists
Page 6, line 9: 2013
Page 6, line 18: abbreviation „BMDCs“ already exists
Page 9, line 9 from the bottom: please explain what is “OVA”
Table 1: species should be reported for all mutations
Figure 1: It is well known that ATP binding site is localized between subunits, therefore P2X monomer apparently cannot bind ATP. According to my opinion, this picture should show trimeric P2X7 because there are no evidence that monomer can be activated with ATP.
Author Response
Reviewer 2
The authors of this review summarized recent literature on the role of P2X7 in immune responses. They focused on the involvement of P2X7 in the antigenic presentation by dendritic cells and macrophages, the activation of T-cells, and the effect of loss or gain of functional polymorphisms on the modulation of immune responses. The authors showed that many P2X7 activities are still waiting to be explained and concluded that the link between the activation of P2X7 and its direct effect on antigen presentation is a promising area of scientific research that may lead to new immunological therapy mediated by the P2X7 receptor. It is a highly informative, actual and excellently written overview that I don't have any great comments on.
Answer: Thank you for the good appreciation of our manuscript.
I only have one suggestion I'm giving authors to consider: recent studies have shown that the P2X7 receptor can be released into the circulation and that the level in the blood increases with various diseases (Giuliani et al., Front Immunol, 2019. 606 10: pp. 793). The authors could comment on possible changes in the level of P2X7 in the blood during inflammatory processes.
Answer: Thank you for your suggestion. Research conducted by Giuliani et al was discussed and added to the manuscript (L337-348)
Minor Comments
Some abbreviations are introduced repeatedly in the text (such as APC and DCs); they should only be mentioned once, when they are used for the first time.
Answer: Thanks for the comment, this was corrected throughout the manuscript.
Page 1, line 8 from the bottom: : „MΦ” is this abbreviation correct? It seems not to be used systematically.
Answer: MΦ (DOI=10.3389/fimmu.2019.01140 ) and Mø (doi:10.1080/1061186X.2020.1775236) can be used. We decided to use “Mø” throughout the entire text.
Page 2, line 4: abbreviation „P2X7R“ has not been introduced and is also not systematically used
Answer: The abbreviation "P2X7R" has been introduced (L42) and used throughout the manuscript.
Page 2, line 6: abbreviation APCs already exists
Answer: This was corrected throughout the manuscript
Page 2, line 18 from the bottom: abbreviation DCs already exists
Answer: This was corrected throughout the manuscript
Page 2, line 8 from the bottom: “Strong evidence suggests that pannexin pores play a critical role in the pore-like activity of P2X7R. “ The authors should mention also other possibilities. More recent studies demonstrated using whole-cell recordings that NMDG+-mediated currents occur without delay after agonist exposure (Harkat, Peverini et al. 2017, Pippel, Stolz et al. 2017), for example. Measurement of the dye uptake activity of purified panda P2X7 receptor reconstituted into liposomes showed that P2X7 forms a dye-permeable pore in the absence of other cellular components but is dependent on the lipid composition of the membrane (Karasawa, Michalski et al. 2017).
Answer: Other pore-opening possibilities were pointed out (L105-115).
Page 3, line 4: “el-Moatassim C and Dubyak GR identified…“ The names of the authors mentioned in the main text should be without abbreviations of their first namest. The same problem is on page 4, line 7 from the bottom and in other places
Answer: Thanks for the observation, this was reformulated, and the names of the authors are no longer indicated, only the surnames; L122-124, L181, L222, L226, L232, L294, L502.
Page 4, line 2: “In addition to its normal functions,,,,“ What is “normal function”? Please, specify.
Answer: Thanks for the observation. This sentence was corrected. We refer to "other non-canonical functions of the P2X7 receptor" (L159)
Page 5, line 8 from the bottom: abbreviation „MHC“ already exists
Answer: This was corrected in the manuscript
Page 6, line 9: 2013
Answer: The reference year on this line (L226) is correct (2023).
Page 6, line 18: abbreviation „BMDCs“ already exists
Answer: This was corrected in the manuscript
Page 9, line 9 from the bottom: please explain what is “OVA”
Answer: This was defined on line 281.
Table 1: species should be reported for all mutations
Answer: Table 1 was corrected according to your indication
Figure 1: It is well known that ATP binding site is localized between subunits, therefore P2X monomer apparently cannot bind ATP. According to my opinion, this picture should show trimeric P2X7 because there are no evidence that monomer can be activated with ATP.
Answer: Thanks for your appreciation. A model proposed by García-Marcos et al. suggests that the P2X7 receptor may act as a non-selective ion channel and lead to pore formation or transduce signals, depending on its location on the plasma membrane. According to this model, the receptor is distributed between lipid rafts and non-raft-associated membrane regions. In the latter, P2X7R can form homotrimers in the presence of ATP and function as an ion channel. However, in lipid rafts, the receptor remains in its monomeric conformation and does not act as a channel but instead activates intracellular signaling pathways (doi:10.1194/jlr.M500408-JLR200). This is described in L144-153. The figure 1 was modified, the homotrimer was included in regions not associated with lipid rafts, and a more extended C-terminal domain was included in the image.

Reviewer 3 Report (New Reviewer)
Comments and Suggestions for Authors
The paper entitled “P2X7 receptor in dendritic cells and macrophages: Implications in antigen presentation and T lymphocyte activation” by Acuña-Castillo et al. is an overview of biology and function of the P2X7 receptor.
This paper is interesting, however the authors may wish to consider the following prior to publication.
- Abstract: I suggest to use "a member of..." instead "a prominent member of...".
- Figure 1: Please modify the figure. P2X7 receptor is a trimer. Authors should highlight that the intracellular domain in the figure is truncated. The P2X7 has a big intracellular bundle putatively endowed of a GTPase activity.
Paragraph 2. Please add few sentences about P2X7 receptor function in biological barriers such as blood brain barrier and blood retinal barrier (please report the following relevant papers: PMID: 30091000, PMID: 31302133). I would also highlight the implication of P2X7 receptor in retinal diseases (please report the following relevant papers: PMID: 35134386; PMID: 28479300; PMID: 35920844).
Please briefly discuss on p2x7 and neuroinflammation (please report the following relevant papers: PMID: 36435235; PMID: 32798466)
Reference Section: please pay attention to the style of the Journal (e.g. several refs have Capitol Letters for the authors names or titles). Please avoid old refs (e.g. #14,#21,#76)
Comments on the Quality of English Language
minor editing
Author Response
Reviewer 3
The paper entitled “P2X7 receptor in dendritic cells and macrophages: Implications in antigen presentation and T lymphocyte activation” by Acuña-Castillo et al. is an overview of biology and function of the P2X7 receptor.
This paper is interesting, however the authors may wish to consider the following prior to publication.
Answer: Thank you for the good appreciation of our manuscript
Abstract: I suggest to use "a member of..." instead "a prominent member of...".
Answer: This was corrected in the manuscript (L18)
Figure 1: Please modify the figure. P2X7 receptor is a trimer. Authors should highlight that the intracellular domain in the figure is truncated. The P2X7 has a big intracellular bundle putatively endowed of a GTPase activity.
Answer: Thanks for your appreciation. The figure was corrected, and now the long intracellular domain corresponding to the C-terminal domain of P2X7R is represented in Figure 1
Paragraph 2. Please add few sentences about P2X7 receptor function in biological barriers such as blood brain barrier and blood retinal barrier (please report the following relevant papers: PMID: 30091000, PMID: 31302133). I would also highlight the implication of P2X7 receptor in retinal diseases (please report the following relevant papers: PMID: 35134386; PMID: 28479300; PMID: 35920844).
Please briefly discuss on p2x7 and neuroinflammation (please report the following relevant papers: PMID: 36435235; PMID: 32798466)
Answer: A brief function of the P2X7 receptor in biological barriers and neuroinflammation was added to the manuscript (L71-78). The highlighted implication of the P2X7 receptor in retinal diseases was not added because it departs from the focus of the paper. The references you suggest were added.
Reference Section: please pay attention to the style of the Journal (e.g. several refs have Capitol Letters for the authors names or titles). Please avoid old refs (e.g. #14,#21,#76)
Answer: According to the Instructions for Authors, Free Format Submission section of the journal, the following is indicated: "Your references may be in any style, provided that you use consistent formatting throughout. It is essential to include author(s) name(s), journal or book title, article or chapter title (where required), year of publication, volume and issue (where appropriate) and pagination. DOI numbers (Digital Object Identifier) are not mandatory but highly encouraged".
On the other hand, the old references included in our work are due to their relevance as primary and fundamental sources in the development of the topic addressed. Despite their age, we consider it essential to grant them appropriate recognition for their pioneering and significant contribution to the initiation and evolution of the field of study described

This manuscript is a resubmission of an earlier submission. The following is a list of the peer review reports and author responses from that submission.
Round 1
Reviewer 1 Report
Comments and Suggestions for Authors
Line 18: "The ATP-gated and cation-selective P2X7 receptor is expressed ..."
- P2X7 receptor creates the non-selective cation channel, so
"The ATP-gated and cation non-selective P2X7 receptor is expressed ...
Author Response
Line 18: "The ATP-gated and cation-selective P2X7 receptor is expressed ..."
- P2X7 receptor creates the non-selective cation channel, so
"The ATP-gated and cation non-selective P2X7 receptor is expressed ...
Answer: We deeply regret the mistake. This was fixed. Also we apologize for the number of errors in the work, these were corrected including errors in definitions.

Reviewer 2 Report
Comments and Suggestions for Authors
Author Response
Acuña-Castillo et al’s paper shows the importance of the of the P2X7 receptor and its function in the immune system. As a non-immunologist I find it a useful review, however, it was difficult for me to follow, because of the lot of abbreviations and not explained phrases (like co-dressing). Maybe it would be worth to add a short definition about the specifical immune phrases to make it more understandable for not immunologist, too. Also, it may be worth to split the paragraph more and make smaller, easier to read parts (the 3rd and 4th paragraph is especially, in their titles are also contain more “things” these could be subdivided).
Answer: The abbreviations and concepts were briefly defined for better understanding. For example, cross-dressing is defined in L254-258
The paragraphs were subdivided, and four figures and one table were added for a better visual understanding of the content. We hope that this new version will be according to your suggestions.
Questions and comments:
line 37: there is a Ca+2 instead of Ca2+ and the + for K and Na are not in supscript
Answer: This was corrected in the manuscript (L72)
line 38: the first reference is the 2 not 1 (and with a fast check I cannot find the 1st
reference in the text)
Answer: We are sorry for the error, the references were checked in the revised version
line 234: M. Tubercolosis should be written Mycobacterium tuberculosis and the -species could not be put there, because it is only 1 species. Or only ycobacteriumspecies could be used. Later: M. Bovis again is not correct. It should be M. bovis.
Answer: Thanks for the comment, this was corrected. M. tuberculosis was used (4.1 section)
line 261: “MTB” bacterium – the B in the abbreviation is not the bacterium?
Answer: Thanks for the observation, this was corrected (4.1 section)
Questions: Is there any studies which investigate the P2XR-pannexin connection and which molecular part interact whit which? Is the modification of the pannexin molecule affecting the P2X7?
Answer: We add information on the interaction of pannexin 1 with P2X7 and how the modification of pannexin-1 generates various effects on the immune response mediated by P2X7 (L95-110).
Other opinions/notes:
For me the experimental conditions were missed when the studies were described (not
everywhere, but in some cases): it is important that a result came from an animal/tissue or cell culture – how precisely e.g. the ATP concentrations could be modulated, how other molecules/cells could affect the results.
Answer: The study model (in vitro, ex vivo, in vivo mouse, or human cell) was added to the described investigations. ATP concentrations were not added to all studies, only when it was necessary to indicate it to describe the receptor effect.
Maybe the P2X7R structure (2nd paragraph) and the differences in the “original” structure (5th paragraph) could be put next to each other.
Answer: Thanks for the comment. However, this should go to the end of the review because some features have to be described earlier in the body of the text
Also more figure could make the manuscript nicer and more easier to follow.
Answer: We added four figures and one table to improve the understanding of the essential events of the P2X7R described in this review.

Reviewer 3 Report
Comments and Suggestions for Authors
This review summarised the literature describing the contribution of P2XR7 receptor activity to antigen presentation and T cell activation.
The main strength of the review was the impressive volume of literature that was included. This makes the review worthwhile as a reference point for other researchers.
There were weaknesses that need addressing.
The structure of the review was very dense and extra figures or tables to summarise information, add information that was unclear (variants were poorly explained), provide visual understanding of mechanism and indeed provide white space would improve the presentation.
The text in different sections did not always appear to be the focus of the section titles. The sentence organisation was haphazard and did not always follow a logical order.
The word activation was used liberally and became confusing. An clear explanation of what is an activated cell and what is meant by activated P2XR7 would be beneficial.
The discussion on each paper was sparse and lacked information that a reader would be looking to find. There was little detail on whether a study was in mouse or man. There was little information on whether studies were in vitro or in vivo, what methods were used to provide evidence or whether the outcomes were conflicting with other studies. How was P2XR7 shown to activate a pathway? How is P2XR7 implicated – is this coincidental data or stronger? Less volume of literature but more critique and discussion on the crucial papers would improve the review.
There were gaps in the literature chosen. Line 77 “Wilhelm K et al. reported in a graft-versus-host disease (GVHD) model that bone marrow DCs (BMDCs) increased the expression of co-stimulatory molecules CD80 and CD86 in vitro and in vivo …”and two other papers pertaining to GVHD were listed. However the Sluyter lab has published extensively on P2XR7 in GVHD and references relating to their work on GVHD was not included. It is important to discuss the models used in these studies. There was similar absence of the work from Zitvogel on variants expressed by APC and their relationship to outcomes in breast cancer.
Comments on the Quality of English LanguageThe written English was of a high grammatical standard. However some of the sentences could be improved by using nouns instead of pronouns and making sure each sentence says what is want to be said.
The sentence starting on L114 begins and ends with similar clauses. This means the sentence does not clearly add information and is circular.
There is an overuse of words such as however, in addition, additionally, interestingly, and furthermore at the start of sentences. This add to the word count whilst not adding to clarity.
Author Response
This review summarized the literature describing the contribution of P2XR7 receptor activity to antigen presentation and T cell activation.
The main strength of the review was the impressive volume of literature that was included. This makes the review worthwhile as a reference point for other researchers.
There were weaknesses that need addressing.
The structure of the review was very dense and extra figures or tables to summarise information, add information that was unclear (variants were poorly explained), provide visual understanding of mechanism and indeed provide white space would improve the presentation.
Answer: The structure of the manuscript was reorganized and separated by theme. We added four figures and one table to summarize and improve your understanding and presentation.
The text in different sections did not always appear to be the focus of the section titles. The sentence organisation was haphazard and did not always follow a logical order.
Answer: The review was deeply reorganized to give a more logical order. We hope that this new version complies with your suggestion.
The word activation was used liberally and became confusing. An clear explanation of what is an activated cell and what is meant by activated P2XR7 would be beneficial.
Answer: The canonical and non-canonical functions of P2X7R were specified. The receptor activation can generate different effects in the cell, which were detailed (section 2).
The discussion on each paper was sparse and lacked information that a reader would be looking to find. There was little detail on whether a study was in mouse or man. There was little information on whether studies were in vitro or in vivo, what methods were used to provide evidence or whether the outcomes were conflicting with other studies. How was P2XR7 shown to activate a pathway? How is P2XR7 implicated – is this coincidental data or stronger? Less volume of literature but more critique and discussion on the crucial papers would improve the review.
Answer: The study models were added, and it was detailed if they were in vitro, ex vivo, or in vivo. The mode of participation of the P2X7 receptor was discussed to explain the described phenomenon. In addition, conflicting studies were pointed out and discussed.
There were gaps in the literature chosen. Line 77 “Wilhelm K et al. reported in a graft-versus-host disease (GVHD) model that bone marrow DCs (BMDCs) increased the expression of co-stimulatory molecules CD80 and CD86 in vitro and in vivo …”and two other papers pertaining to GVHD were listed. However the Sluyter lab has published extensively on P2XR7 in GVHD and references relating to their work on GVHD was not included. It is important to discuss the models used in these studies. There was similar absence of the work from Zitvogel on variants expressed by APC and their relationship to outcomes in breast cancer.
Answer: The work of the Sluyter group at GVDH was incorporated (L276). However, Zitvogel's work on cancer was not discussed, as he either escaped the focus of the main topic or was indirect evidence.
